# Ang II-Induced Hypertension Exacerbates the Pathogenesis of Tuberculosis

**DOI:** 10.3390/cells10092478

**Published:** 2021-09-19

**Authors:** Soo-Na Cho, Ji-Ae Choi, Junghwan Lee, Sang-Hun Son, Seong-Ahn Lee, Tam-Doan Nguyen, Song-Yi Choi, Chang-Hwa Song

**Affiliations:** 1Department of Microbiology, College of Medicine, Chungnam National University, Daejeon 35015, Korea; soona3831@gmail.com (S.-N.C.); jiae9035@gmail.com (J.-A.C.); asrai1509@gmail.com (J.L.); scssh3487@gmail.com (S.-H.S.); ahn7773@gmail.com (S.-A.L.); ntdoan95@gmail.com (T.-D.N.); 2Department of Medical Science, College of Medicine, Chungnam National University, Daejeon 35015, Korea; 3Department of Pathology, College of Medicine, Chungnam National University, Daejeon 35015, Korea; sychoi@cnu.ac.kr; 4Translational Immunology Institute, Chungnam National University, Daejeon 34134, Korea

**Keywords:** *Mycobacterium tuberculosis*, Angiotensin II, hypertension, foamy macrophages

## Abstract

It has been known that infection plays a role in the development of hypertension. However, the role of hypertension in the progression of infectious diseases remain unknown. Many countries with high rates of hypertension show geographical overlaps with those showing high incidence rates of tuberculosis (TB). To explore the role of hypertension in tuberculosis, we compared the effects of hypertension during mycobacterial infection, we infected both hypertensive Angiotensin II (Ang II) and control mice with *Mycobacterium tuberculosis* (*Mtb*) strain H37Ra by intratracheal injection. Ang II-induced hypertension promotes cell death through both apoptosis and necrosis in *Mtb* H37Ra infected mouse lungs. Interestingly, we found that lipid accumulation in pulmonary tissues was significantly increased in the hypertension group compared to the normal controls. Ang II-induced hypertension increases the formation of foamy macrophages during *Mtb* infection and it leads to cell death. Moreover, the hypertension group showed more severe granuloma formation and fibrotic lesions in comparison with the control group. Finally, we observed that the total number of mycobacteria was increased in the lungs in the hypertension group compared to the normal controls. Taken together, these results suggest that hypertension increases intracellular survival of *Mtb* through formation of foamy macrophages, resulting in severe pathogenesis of TB.

## 1. Introduction

Tuberculosis (TB) is an ancient and active disease that has afflicted humankind for more than 4000 years [1]. TB is caused by *Mycobacterium tuberculosis* (*Mtb*), for which humans are the only known reservoir [2]. Many countries with high rates of hypertension show geographical overlaps with those showing high incidence rates of TB [3]. A previous report suggested that pulmonary hypertension contributes to poor prognosis of pulmonary TB [4]. However, there have been no reports regarding how hypertension affects mycobacterial infection. Hypertension is a risk factor for cardiovascular disease (CVD) and is responsible for large numbers of deaths by heart disease, stroke, and myocardial infarction worldwide [5,6]. The levels of C-reactive protein (CRP) and inflammatory cytokines, such as tumor necrosis factor-α (TNF-α) and interleukin-6 (IL-6), are increased in hypertensive patients [7,8]. Angiotensin II (Ang II), which causes vasoconstriction resulting in high blood pressure, regulates the vascular tone and synthesis of proinflammatory cytokines [9]. Ang II receptor blockers reduce the circulating levels of some inflammatory mediators, such as IL-6, TNF-α, monocyte chemoattractant protein 1 (MCP-1), and CRP, in the vessels [10,11,12,13]. Bacterial infections have been shown to be associated with atherosclerosis and CVD [14,15]. Some reports have indicated that a variety of other infectious agents are involved in the pathogenesis of atherosclerosis [16,17]. These reports suggest that bacterial infections are closely related to hypertension.

A previous report suggested that risk of cardiovascular disease in persons who develop tuberculosis is higher than in persons without a history of tuberculosis [18]. A recent paper showed that tuberculosis appears to be a marker for increased CVD risk [19]. Chronic infection with several viruses and bacteria has been reported in association with high blood pressure [20]. Interestingly, hypertension is reported to be associated with the severity and fatality of SARS-CoV-2 infection [21]. In this study, we hypothesized that hypertension plays a critical role in the severe pathogenesis of TB.

Lung inflammation might contribute to the development of acute CVD [22]. Inflammatory reaction is associated with not only a salutary response but also apoptosis of immune cells to injurious events [23]. Cell death can trigger inflammatory responses [24]. Apoptosis of *Mtb*-infected macrophages is directly associated with mycobacterial killing, whereas necrosis allows the release of viable mycobacteria for subsequent re-infection [25,26]. Although it is widely accepted that apoptosis of *Mtb*-infected macrophages is beneficial to host defense and necrosis is a mechanism for transmission of infections, there is still much to be studied. *Mtb* induces inflammation in the infected organ and the persistent inflammatory response leads to damage to the target organ [27]. The survival mechanisms of *Mtb* involve triggering an anti-inflammatory response and blocking the production of reactive oxygen species and nitrogen intermediates, etc. [28,29]. Both the balance of pro- and anti-inflammatory cytokines in relation to the stage of *Mtb* infection are important for disease progression [30,31]. The cell-mediated immune response to *Mtb* affects the coronary vessels, inducing inflammation and resulting in atherogenesis [18]. Recent studies reported clinical cases of sudden cardiac death associated with tuberculous myocarditis [18,32].

In the present study, we hypothesized that hypertension may be harmful to the host defense system against mycobacteria. To determine the role of hypertension in *Mtb* infection, we analyzed the effects of Ang II on suppression of intracellular survival of *Mtb*.

## 2. Materials and Methods

### 2.1. Ethics Statement

Male C57BL/6 mice were purchased from Samtako Company (Gyeonggi-do, Korea). All animal experiments were approved by the Ethics Committee and Institutional Animal Care and Use Committee of the Laboratory Animal Research Center at Chungman National University Medical School (Daejeon, Korea, CNU-00907). All animal studies were performed in accordance with the guidelines of the Korea Food and Drug Administration (KFDA).

### 2.2. Hypertensive Animal Model and Blood Pressure Measurement

We used 8-week-old male C57BL/6 mice in this study. Mice were anesthetized by intraperitoneal injection of tribromoethanol (Avertin, 1.25 mg/kg; Acros Organics, Morris Plains, NJ, USA). Angiotensin II (Ang II, 0.7 mg·kg^−1^·day^−1^; Calbiochem, San Diego, CA, USA) was infused into C57BL/6 mice using an implanted Alzet micro-osmotic pump (Durect, Cupertino, CA, USA) for 14 or 28 days. After 1 week of Ang II infusion, animals were infected by intratracheal injection of 1 × 10^6^/CFU of *Mtb* H37Ra (ATCC 25177). Systolic blood pressure measurements were performed in trained conscious mice once a week by the tail-cuff method, and more than 10 recordings were averaged (Lab Chart 8 reader system; Colorado Springs, CO, USA). The *Mtb*-infected mice were sacrificed, and the heart, aorta, and lung tissues were obtained at 4 weeks after Ang II infusion.

### 2.3. Histopathological Analysis and Scanning Microscopy

C57BL/6 mice were perfused with 4% paraformaldehyde (PFA) wash for 5 min or until lungs are cleared of blood. The paraffin-embedded tissues were cut into sections 3–5 μm thick on a microtome. Tissue samples on slides were deparaffinized with xylene and then rehydrated through a graded alcohol series. The tissue slides were stained with H&E, Masson’s trichrome stain, and immunohistochemistry (IHC), and assessed for severity of cell death or fibrosis. Intracellular survival of *Mtb* was determined using AFB staining of the lung tissue slides.

### 2.4. Hematoxylin and Eosin Stain

The slides were incubated with Gill’s V Hematoxylin (Muto Pure Chemicals Co., Tokyo, Japan) for 5 min, washed with running tap water for 5 min, decolorized with 0.3% HCl solution for 15 s, and then rinsed with Scott’s Bluing solution for 7 min. The cytoplasm was stained with Eosin Y (Muto Pure Chemicals) solution for 2 min, and then the slides were dehydrated, cleared, and mounted.

### 2.5. Acid-Fast Bacillus Stain

Lung tissue section was stained to visualize the bacteria with AFB (Ziehl–Neelsen) staining, slides were incubated with heated carbol-fuchsin solution for 5 min, washed with running tap water for 5 min, decolorized with 1% acid alcohol until light pink and color stops, washed with running tap water for 5 min, and then rinsed with distilled water, working contrast stained by methylene blue for 2 min, and rinsed with distilled water.

### 2.6. Immunohistochemistry (IHC)

The slides were deparaffinized and pretreated for heat-induced antigen retrieval with citrate buffer (pH 6.0). After blocking of endogenous peroxidase activity with 3% hydrogen peroxide (H_2_O_2_ and 1% bovine serum albumin; BSA), tissue sections were incubated with the indicated antibodies overnight at 4 °C. IHC was performed with antibodies against Cleaved caspase-3 (c-Cas3, 1:100; Cell Signaling Technology), Hypoxia induced factor 1α (HIF-1α, 1:100; Santa Cruz Biotechnology, Santa Cruz, CA), and lactate dehydrogenase A (LDH-A, 1:100; Abcam, Cambridge, MA) using an LSAB2 system horseradish peroxidase (HRP) kit (Dako, Carpinteria, CA). A biotinylated secondary antibody was added, followed by incubation with Vectastain ABC reagent (Vector Laboratories, California, USA) followed by treatment with 3,3′-diaminobenzidine (DAB), peroxidase substrate until the desired staining intensity had developed.

### 2.7. Oil Red O Stain

The lung tissue was placed into graded sucrose (10%–20%–30%) solutions for overnight incubation at 4 °C. The tissues were frozen in liquid nitrogen and cut into sections 30 μm thick on a cryotome. The tissue sections were placed on slides, air dried, washed with running tap water, and rinsed with 60% isopropanol. BMDM cells were fixed with 4% PFA on ice for 30 min followed by washing with running 1x PBS and rinsing with 60% isopropanol. Cells were then stained with freshly prepared Oil Red O solution for 30 min at room temperature and subjected to nuclear staining with Gill’s V Hematoxylin.

### 2.8. Histological Analysis

The tissue sections were counterstained, mounted, and visualized under an optical microscope (Olympus BX51; Olympus, Tokyo, Japan) or ScanScope CS system (Aperio Technologies, Vista, CA, USA). For analysis of granuloma formation and inflammation, stained lung sections were photographed using a microscope fitted with a camera (ScanScope CS system) that was connected to a computer.

### 2.9. Cell Culture and Bacterial Infection

Murine macrophage Raw 264.7 cells were maintained in Dulbecco’s modified Eagle’s medium (DMEM) supplemented with 10% FBS, penicillin (100 IU/mL), and streptomycin (100 µg/mL). The cells (1 × 10^5^) were cultured in 12-well polypropylene tissue culture plates overnight at 37 °C, 5% CO_2_ to allow cell adherence before infection. Bone marrow derived macrophage cells (BMDMs) were isolated from femurs and tibias of C57BL/6 mice (6–8 weeks old) and then differentiated by growth for 4–5 days in medium containing macrophage colony-stimulating factor (M-CSF, 25 µg/mL; R&D Systems, Minneapolis, MN, USA). The cells were infected for 3–5 h with *Mtb* H37Ra (ATCC 25177). Then, cells were washed to remove excess bacteria and cultured with fresh complete medium without antibiotics.

### 2.10. Cytotoxicity Analysis

Cytotoxicity analysis was conducted using an Annexin V/Propidium iodide (PI) staining kit (BD Pharmingen, San Diego, CA, USA). The cells were stained with FITC-conjugated Annexin V and PI. Analysis of the stained cells was performed on a FACS Canto II system (BD Biosciences, North Brunswick, NJ, USA), and the results were analyzed using Flow Jo V10 software (Tree Star, Ashland, OR, USA).

### 2.11. FACS Analysis for CD36 Surface Expression

BMDM cells were pretreated with Ang II (1 μM) and then incubated with oxidized low-density lipoprotein (OxLDL, 50 μg/mL) after H37Ra (multiplicity of infection, MOI = 1) infection for 24 h. After infection, cells were harvested with trypsin and washed twice in PBS. BMDM cells were stained with APC-conjugated anti-CD36 antibody (1:100) for 30 min at room temperature. After washing three times in 1× PBS, the cells were resuspended in 500 μL of FACS flow buffer and analyzed on a FACS Canto II system (BD Biosciences). The results were analyzed using Flow Jo V10 software (Tree Star).

### 2.12. LDH Cytotoxicity Detection

Raw 264.7 cells were pretreated with 1 μM Ang II for 1 h followed by incubation with 50 μg/mL OxLDL for 48 h after infection with *Mtb*. Necrotic cell death was quantified by measurement of LDH released from damaged or disrupted cells using an LDH cytotoxicity detection kit (Takara Korea Biomedical, Seoul, Korea) in culture medium. Released LDH was assayed by determining the absorbance at 450 nm (A450) using a microplate reader (SpectraMax^®^ ABS; Molecular Devices, Downingtown, PA, USA).

### 2.13. Western Blotting Analysis

Mouse macrophage cells or tissue lysates were incubated with radio-immunoprecipitation assay buffer (150 mM NaCl, 1% Triton X-100, 1% sodium deoxycholate, 50 mM Tris-HCl, pH 7.5, 2 mM EDTA, 5 mM NaF, and 5 mM Na3VO4) containing protease inhibitors (ELPIS Biotech, Daejeon, Korea). Proteins were separated by 10–15% SDS-polyacrylamide (SDS-PAGE), and then electroblotted onto polyvinylidene difluoride membranes, blocked, and incubated overnight with primary antibodies. Antibody binding was visualized using the appropriate HRP-conjugated or alkaline phosphatase-conjugated secondary antibodies (anti-mouse IgG; Calbiochem and anti-rabbit IgG; Cell Signaling Technology) with enhanced chemiluminescent substrate (EMD Millipore, Billerica, MA, USA). Band intensities were quantified using Omega Lum C (Aplegen Inc., Pleasanton, CA, USA). The primary antibodies used in the present study were anti-Caspase-3 (1:1000; Cell Signaling Technology) and anti-β-actin (1:2000; Santa Cruz Biotechnology).

### 2.14. Intracellular Survival Assay

The intracellular survival of *Mtb* H37Ra in C57BL/6 mice and BMDM cells was quantified by analysis of colony forming units (CFU). Mice were anesthetized and infected by intratracheal injection of 1 × 10^6^/CFU of *Mtb* H37Ra. At 21 days after *Mtb* infection, the mice were sacrificed, and the lungs were isolated. BMDM cells were infected with *Mtb* H37Ra (MOI = 1) after treatment with Ang II (1 μM, 1 h), and incubated for 3 h at 37 °C in 5% CO_2_. The cells were washed three times with PBS to remove extracellular bacteria and then incubated with 50 μg/mL OxLDL for the indicated times. The lung tissues and BMDM cells were lysed with autoclaved distilled water to harvest the intracellular bacteria. The lysates were plated on Middlebrook 7H10 agar plates and incubated at 37 °C for 14–21 days. Colony counts were performed in triplicate.

### 2.15. Statistical Analysis

Statistical analysis was performed using one-way analysis of variance (ANOVA) with Bonferroni post hoc test using GraphPad Prism version 5.0 (GraphPad Software, San Diego, CA, USA). Data are presented as means ± SD or SEM. In all analyses, *p* < 0.05 was taken to indicate statistical significance.

## 3. Results

### 3.1. Ang II-Induced Hypertension Increases the Cell Death in Mtb Infected Mice Lungs

To examine the effect of hypertension on *Mtb* infection, we established a hypertensive mouse model using Ang II. Hypertension was analyzed by measuring the enhanced systolic blood pressure of mice using the tail cuff method on a weekly basis. Systolic blood pressure in Ang II infused mice was (180 ± 10 mmHg) significantly increased at 1 week after infusion compared to the control (120 ± 10 mmHg). The increased systolic blood pressure by Ang II was not affected by *Mtb* infection (Figure 1a). The elevated blood pressure in the Ang II-stimulated group was significantly decreased at 4 weeks after cessation of Ang II infusion for 2 weeks (Figure 1a).

To examine the effect of hypertension on granuloma formation, we measured the inflammatory response and fibrosis in mice lungs during *Mtb* infection. hematoxylin and eosin (H&E) staining for histopathological analysis (upper) and Masson’s trichrome staining for the detection of collagen fibers (bottom) was performed in mice lungs (Figure 1b). Interestingly, severe tissue damage and pulmonary fibrosis were observed in the lung tissue of *Mtb*-infected hypertensive mice (Ang II infusion for 4 weeks) in comparison with the other groups. The aggregation of lymphocytes intermixed with histiocytes was found along with the alveolar spaces and the inflammatory reaction was strongly induced in the lungs of Ang II-infused mice compared with the control group (Figure 1b). Abundant infiltration of inflammatory cells, including histiocytes, was identified along the alveolar walls and peribronchial spaces in the lung tissue of *Mtb*-infected hypertensive mice (Ang II infusion for 4 weeks) (Figure 1b). The lungs of *Mtb*-infected hypertensive mice showed less fibrotic severity, inflammatory reaction, and granuloma formation after the termination of Ang II infusion for 2 weeks compared to the 4-week Ang II-infused mice (Figure 1b). These results suggested that Ang II-induced hypertension plays an important role in increasing the inflammatory response with granuloma formation during *Mtb* H37Ra infection.

Apoptosis of *Mtb*-infected macrophages is associated with diminished pathogen viability [26,33]. In contrast, necrosis is necessary to facilitate the dissemination of mycobacteria [25,33]. Previous reports suggested that ER stress plays an important role in Mtb infected macrophages and ER stress contributes to the pathogenesis of hypertension [34,35,36]. Next, we investigated whether high blood pressure affects the death of *Mtb*-infected macrophages because our group has discovered that apoptosis plays an important role in the destruction of tuberculosis bacteria in macrophages [35]. We compared cleaved caspase-3 (Cas-3) in lung tissues from Ang II-induced hypertensive mice with or without Ra-infection using immunohistochemistry (IHC) staining and Western blot analysis. (Figure 2a,b). The production of cleaved Cas-3 was elevated in lung tissue of Ang II-induced hypertensive mice comparing to control during *Mtb* infection. The production of cleaved Cas-3 was elevated in lung tissue of Mtb infected mice comparing to control mice. Ang II-induced hypertension enhances the cleaved Cas-3 more than control during Mtb infection (Figure 2a,b).

Next, we compared the production of LDH-A in mouse lungs because measuring LDH-A release is a useful method for detection of necrosis [37]. The level of LDH-A production was significantly increased in the lungs of Ang II-induced hypertensive mice compared with the control during *Mtb* infection (Figure 3a). Consistent with these findings, we detected the elevated levels of LDH-A in the lungs of Ang II-induced hypertensive mice by immunofluorescence staining (Figure 3b).

Hypoxia is one of the stresses *Mtb* encounters in the granuloma [38] and HIF-1α can mediate hypoxia-induced LDH-A expression [39]. Interestingly, we found the elevated levels of HIF-1α in the lungs of Ang II-induced hypertensive mice (Figure 3c). Ang II stimulation increases LDH-A and HIF-1α production in the lungs of *Mtb*-infected mice compared to the only *Mtb*-infected group. The increased levels of LDH-A and HIF-1α production were reduced by subsequent cessation of Ang II infusion for 2 weeks (Figure 3).

Moreover, we observed that pretreatment of HIF-1α modulator (FM19G11) suppressed LDH-A production at 24 h in Ang II stimulated macrophages during *Mtb* infection in a dose-dependent manner (Appendix A). Our results show that Ang II-induced hypertension promotes cell death through both apoptosis and necrosis in *Mtb* H37Ra infected mouse lungs.

### 3.2. Ang II-Induced Hypertension Enhances the Accumulation of Intracellular Lipids by Increasing the Expression of CD36 on the Macrophage Surface during Mtb H37Ra Infection

A previous report suggested that Mtb infection promotes alteration of macrophages to foam cells characterized by high lipid content [40]. To examine whether Ang II infusion induced lipid accumulation in the lung tissues of Mtb H37Ra-infected mice, we examined lipid accumulation in macrophages by Oil Red O staining (Figure 4a). Distinct enhancement of lipid accumulation was observed in the lungs of Ang II-induced hypertensive mice (after 4 weeks of Ang II infusion) compared to the controls during Mtb infection. Interestingly, the foamy macrophages were abundant in the granuloma like lesions formed by Mtb H37Ra infection. The amount of accumulated lipid was reduced in the hypertensive mice after cessation of Ang II infusion after 2 weeks (Figure 4a). Our data showed that Ang II-induced hypertension increases the formation of foamy macrophages in mouse lungs during Mtb infection.

To examine whether hypertension induces the level of oxidized low-density lipoprotein (OxLDL) in vivo, we compared the concentration of OxLDL in sera between Ang II-induced hypertensive mice group and the control by ELISA. We found that the level of OxLDL was significantly increased in Ang II-induced hypertensive mice compared to the control (Figure 4b). The elevated level of OxLDL in Ang II-induced hypertensive mice was significantly reduced by the cessation of Ang II infusion in Mtb-infected mice (Figure 4b). Next, we examined whether Ang II-induced hypertension increases expression of scavenger receptors (CD36) associated with the ability to uptake lipids (OxLDL) in macrophages. We found that Ang II stimulation significantly increased the accumulation of OxLDL in the cytosol of bone-marrow-derived macrophages (BMDMs) during Mtb H37Ra infection in comparison to unstimulated cells infected with Mtb H37Ra alone (Figure 4c). To investigate the role of CD36 in the formation of foamy macrophages by Ang II during Mtb infection, the levels of CD36 expression on the macrophage surface were measured. By monitoring the uptake of OxLDL, we found that the expression of CD36, was increased by Ang II stimulation in Mtb H37Ra-infected macrophages (Figure 4d). We found that OxLDL accumulation induces foamy macrophages through CD36 during Mtb infection.

Next, we investigated the relationship between Ang II-induced foamy macrophages and cell death during Mtb infection. OxLDL-accumulated Raw 264.7 cells were infected with Mtb in the presence or absence of Ang II for 24 h and then cell death was assessed using Annexin V/PI staining. Compared with the control groups, the apoptosis ratio of Mtb-infected groups was significantly increased in the presence of Ang II (Figure 5a). Consistently, the levels of necrotic cell death by Ang II were significantly enhanced in Mtb-infected foamy macrophages compared with the control groups (Figure 5b). The release of LDH was significantly increased by Ang II in Mtb-infected foamy macrophages comparing with the control groups (Figure 5c). These results showed that Ang II-induced hypertension contributes to induction of foamy macrophages and it sequentially leads to increment of macrophage cell death during Mtb infection.

### 3.3. Ang II-Induced Hypertension Contributes to Intracellular Survival of Mtb in the Mice Lungs

Previous reports suggested that foamy macrophages give an excuse for survival and maintenance of Mtb [41]. Next, we investigated whether Ang II-induced hypertension increases the dissemination and survival of Mtb in foamy macrophages, the total number of intracellular mycobacteria was enumerated in the lung tissues from mice following infection with Mtb after 2 or 4 weeks of Ang II infusion using acid-fast bacillus (AFB) staining (Figure 6a) and CFU assay (Figure 6b). Unexpectedly, Mtb growth was significantly increased in the lungs of mice after 2 and 4 weeks of Ang II infusion. The increase in the number of bacilli was reduced by ceasing Ang II release (Figure 6a,b). Mtb growth was also significantly increased in Ang II-mediated foamy macrophages in comparison to unstimulated cells infected with Mtb H37Ra alone (Figure 6c). Our results suggested that Ang II-induced hypertension contributes to increment intracellular survival of Mtb through the formation of foamy macrophages.

## 4. Discussion

Previous studies have suggested that bacterial infection induces hypertension [16,18]. In the present study, we showed that granuloma like formation, fibrosis, and severe inflammatory lesions were increased in the lungs of Ang II-induced hypertensive *Mtb* H37Ra-infected mice compared to the lungs of unstimulated mice only infected with *Mtb* H37Ra. Hypertension causes severe pathologies and enhanced intracellular survival of bacilli during mycobacterial infection, consistent with our hypothesis. Hypertension increases the expression of adhesion molecules on the vascular surface, leading to infiltration of circulating monocytes and lymphocytes [42]. These immune cells produce inflammatory cytokines, which sequentially lead to lung tissue disruption via granuloma formation and fibrosis [43,44,45]. Hypertension promotes the development of myocardial dysfunction and fibrosis [44]. *Mtb* infection leads to an increased inflammatory response and induces pulmonary fibrosis through granuloma formation [46].

In the present study, we observed increased accumulation of foamy macrophages in the lungs of Ang II-induced hypertensive mice during *Mtb* infection. Our results suggested that Ang II stimulation increased CD36 expression and OxLDL uptake in macrophages, resulting in enhanced intracellular survival of mycobacteria (Figure 7). Previous studies have shown that macrophages take up OxLDL through CD36 or the other scavenger receptor A (SR-A), leading to foamy macrophage formation [47,48]. In addition, other reports have suggested that LDL is more susceptible to oxidation in hypertensive patients [49]. Foamy macrophages are key players in both sustaining bacterial infection and contributing to cavitation in tissues and release of infectious bacilli [50]. The foamy macrophages are abundant in pulmonary granulomas in human TB patients and it seems to sustain intracellular *Mtb* [50]. Ang II-mediated foamy macrophages might provide a favorable environment for dissemination of *Mtb* in the host.

LDH is a well-known biomarker to identify necrotic cells [51] and a previous study showed that LDH production was increased in the sera of patients with infectious lung diseases, such as pneumonia and pulmonary TB [52]. It is well known that necrosis helps in dissemination of Mtb [26,53]. In the present study, we found that *Mtb*-induced LDH production was significantly increased in Ang II-induced hypertensive mice in comparison to controls. Ang II is known to induce HIF-1α production [54] and HIF-1/2α protein promotes LDH-A production via binding to the LDH-A promoter region under hypoxic conditions [39]. Therefore, our results suggested that hypertension induces HIF-1α production, resulting in LDH-A production during *Mtb* infection.

Apoptosis is one of the host defense strategies against *Mtb* infection because it can regulate *Mtb* replication through activating the innate immune response [26]. In contrast, necrosis is necessary to facilitate the dissemination of mycobacteria by disrupting the lung structure and allowing access to the major airways [26,33]. Recent studies have suggested that virulent *Mtb* induces necrosis or inhibits apoptosis in macrophages via impairing plasma membrane repair [55,56]. Our data showed that apoptosis and necrosis were significantly induced in Ang II-stimulated foamy macrophages during *Mtb* infection. These observations suggest that lipid-loaded foamy macrophages under hypertensive conditions may be useful for *Mtb* to survive even though apoptosis was significantly increased in *Mtb*-infected macrophages. Based on previous reports, we assume that *Mtb* used lipids in foamy macrophages for intracellular bacterial growth [57]. A previous report suggested that lipid-loaded foamy macrophages are like the M2 phenotype of macrophages, which may be permissive for *Mtb* survival [57,58]. We have a limitation that the experiment was conducted only in mice, not humans, and the increased foamy macrophages cannot explain all of the correlation between hypertension and TB. Our system cannot distinguish between Ang II direct effects or Ang II-mediated hypertensive effects, but our data suggested that Ang II-induced hypertension contributes to the intracellular survival of *Mtb* through induction of cell death via foamy macrophage formation. Ang II-induced hypertension might be associated with the severity of TB pathogenesis. Therefore, controlling blood pressure is important to reduce severe pathologies of mycobacterial infection.

## 5. Conclusions

In summary, this research provided the evidence that Ang II-induced hypertension contributes to the intracellular survival of *Mtb* through induction of cell death. Our results suggested that lipid-loaded foamy macrophages could be useful for *Mtb* survival in hypertensive conditions. Although there are various causes of hypertension, this study uses only the Ang II-mediated hypertensive model and has a limitation that the in vitro study cannot replace the in vivo studies. Nevertheless, our results may suggest that AngII-induced hypertension may exacerbate pathologies of mycobacterial infection.

## Figures and Tables

**Figure 1 cells-10-02478-f001:**
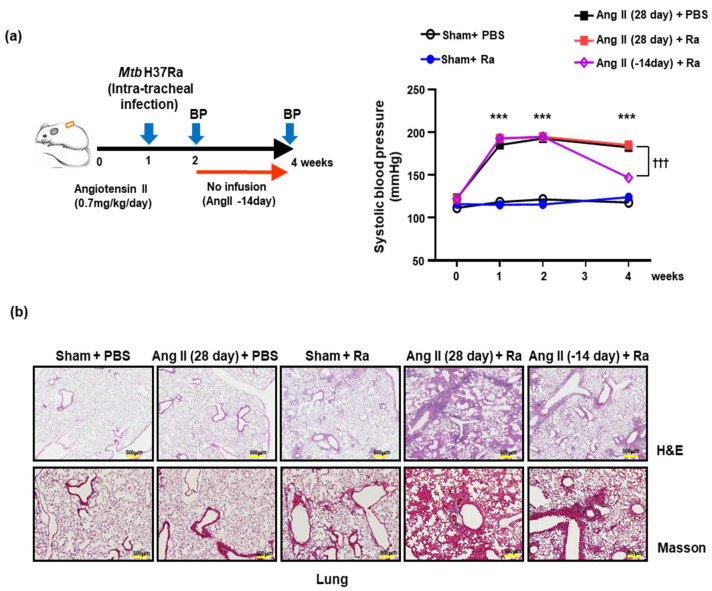
Ang II-induced hypertension promotes tissue damage due to severe inflammation in mice during *Mtb* 37Ra-infection. (**a**) Schematic diagram of the experimental schedule (Left). Angiotensin II (Ang II, 0.7 mg/kg/day) or PBS (Sham) were infused for 28 or 14 days (-14 day, consume the release of Ang II for 2 weeks) using an osmotic mini-pump. After 7 days of Ang II infusion, mice were infected by intratracheal injection of 1 × 10^6^ CFU of *Mtb* H37Ra. Systolic blood pressure was measured once a week by the tail-cuff method (>10 recordings were averaged). Data shown are the means ± SEM of five mice in each group. Data are representative of three independent experiments (^†††^ *p* < 0.001, *** *p* < 0.001). (**b**) Masson’s trichrome (magnification, 20×) and H&E (magnification, 10×) staining were performed after 28 days of Ang II infusion to assess fibrosis (collagen fibers, blue) and granuloma formation in *Mtb* H37Ra-infected hypertensive mice (scale bars = 500 μm). Data are presented as representative pictures of each group.

**Figure 2 cells-10-02478-f002:**
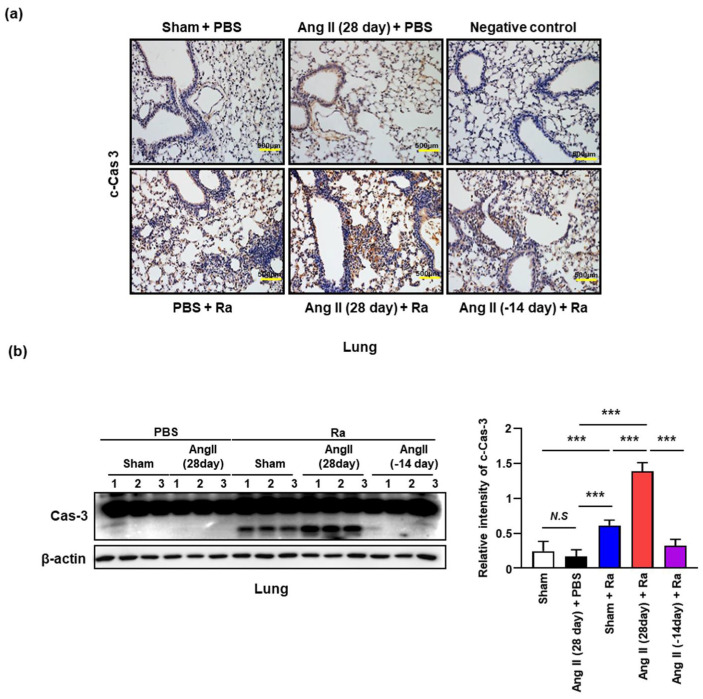
Ang II-induced hypertension increases apoptotic cell death in *Mtb* H37Ra infected mice. (**a**) Ang II (0.7 mg/kg/day) or PBS were infused for 28 or 14 days using an osmotic mini-pump. After 7 days of Ang II infusion, mice were infected by intratracheal injection of 1 × 10^6^ CFU of *Mtb* H37Ra. Mouse lung tissue samples were obtained 3 weeks after infection. IHC was performed to assess the apoptotic cell death with antibodies against cleaved caspase-3 in lung tissues (magnification, 20×). Data are presented as representative pictures of each group. (**b**) Western blotting analysis of lung tissues was performed using appropriate antibodies (Cas-3 and β-actin). Data are representative of three independent experiments. Data are presented as the means ± SD from three experiments (*** *p* < 0.001).

**Figure 3 cells-10-02478-f003:**
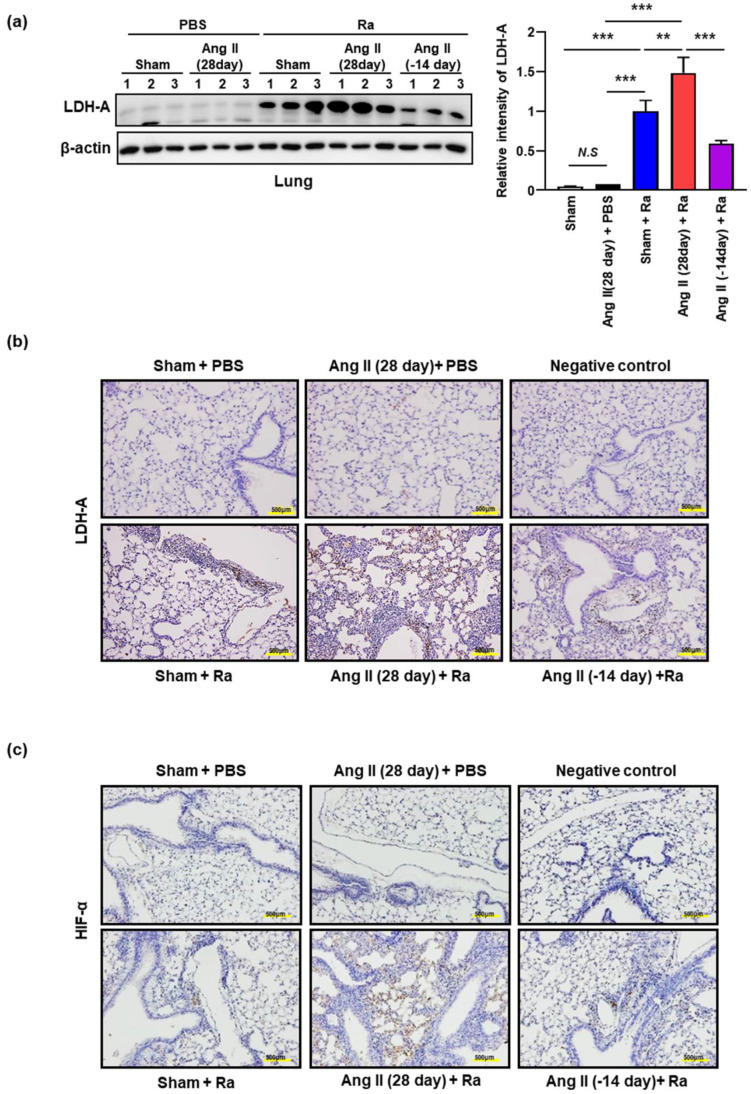
Hypertensive condition by Ang II-infusion promotes necrotic cell death during *Mtb* infection. Hypertension was induced in C57BL/6 mice by Ang II (0.7 mg/kg/day) infusion using an osmotic mini-pump for 14 or 28 days. PBS was used in the control group. Mice were infected with *Mtb* H37Ra (1 × 10^6^ CFU) by intratracheal injection at after 1 week of Ang II infusion. (**a**) Western blotting analysis was performed to examine the expression levels of LDH-A protein after 4 weeks of infusion with Ang II or PBS in *Mtb*-infected mouse lung tissues. Data are representative of three independent experiments. Data are presented as the means ± SD from three experiments (** *p* < 0.01, *** *p* < 0.001). (**b**,**c**) Mouse lung tissues were stained with antibodies to LDH-A and HIF1-α by IHC analysis at 3 weeks after infection (magnification, 20×). Data are presented as representative pictures of each group.

**Figure 4 cells-10-02478-f004:**
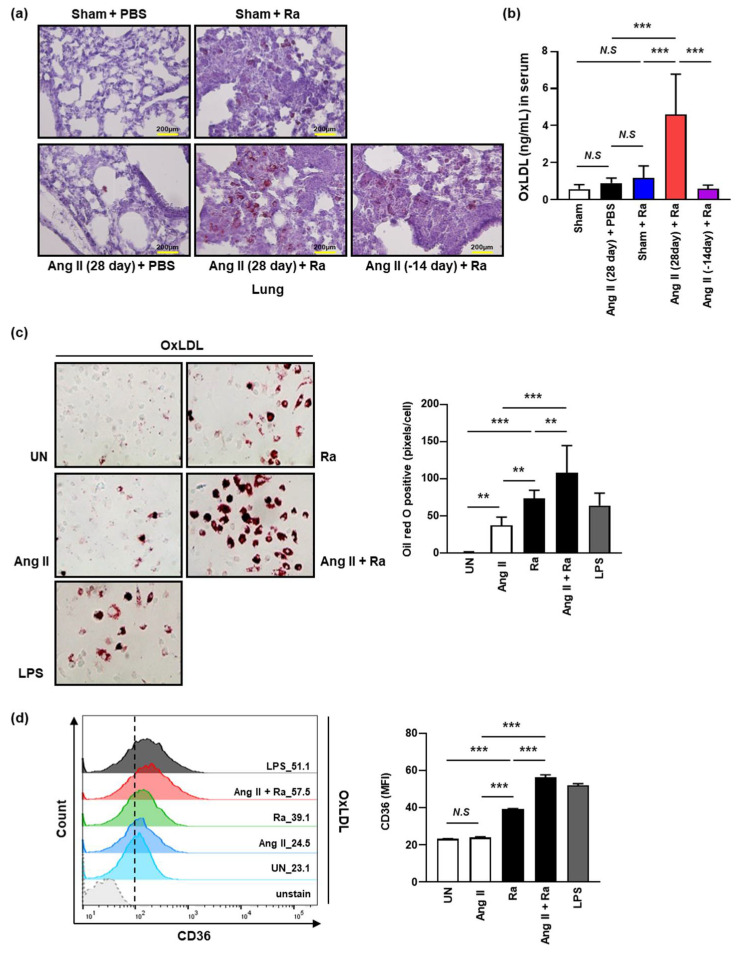
Ang II-induced hypertension provides increasing lipid accumulation during *Mtb* H37Ra infection in macrophages. (**a**,**b**) Hypertension was induced in C57BL/6 mice by Ang II (0.7 mg/kg/day) infusion. After 7 days of Ang II infusion, mice were infected by intratracheal injection of 1 × 10^6^ CFU of *Mtb* H37Ra. (**a**) Oil Red O staining for detection of lipid (red) in lung tissue derived from *Mtb* H37Ra-infected or uninfected mice in the Ang II-induced hypertension and normal control groups (magnification, 40×). Data are presented as representative pictures of each group. (**b**) The OxLDL content was detected by the OxLDL ELISA kit in mice sera at 3 weeks after infection. Released OxLDL in the mouse sera was measured by determining the absorbance at 450 nm (A450). (**c**,**d**) BMDM cells were pretreated with Ang II (1 μM) for 1 h followed by incubation with 50 μg/mL OxLDL after *Mtb* H37Ra (MOI = 1) infection. (**c**) Oil Red O staining was used for detection of foamy macrophage formation at 48 h after infection. LPS (500 ng/mL, 24 h) was used as a positive control. (**d**) BMDM cells were incubated with APC-conjugated anti-CD36 antibody (1:100) at room temperature. After washing, the cells were analyzed by flow cytometry. Data are presented as the means ± SD from three experiments. Data are representative of three independent experiments (** *p* < 0.01, *** *p* < 0.001).

**Figure 5 cells-10-02478-f005:**
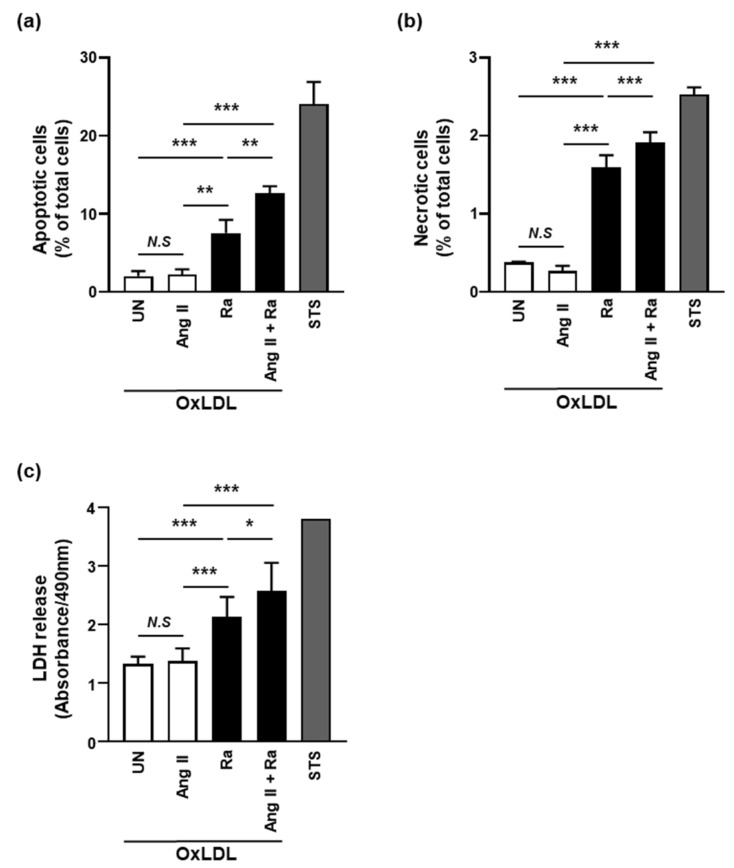
OxLDL accumulation provides a favorable environment for the growth of mycobacteria in Ang II-stimulated macrophages. (**a**–**c**) Raw 264.7 cells were pretreated with Ang II (1 μM) for 1 h followed by incubation with 25 μg/mL OxLDL after Mtb H37Ra (MOI = 1) infection. (**a**,**b**) Cell death was examined by FACS analysis with annexin V/PI staining at 24 h after Mtb infection. Staurosporine (STS, 500 nM, 6 h) was used as a positive control for cell death. (**c**) Released LDH in the culture supernatant was measured by determining the absorbance at 490 nm (A490) at 24 h after infection. Data are representative of three independent experiments (* *p* < 0.05, ** *p* < 0.01, *** *p* < 0.005).

**Figure 6 cells-10-02478-f006:**
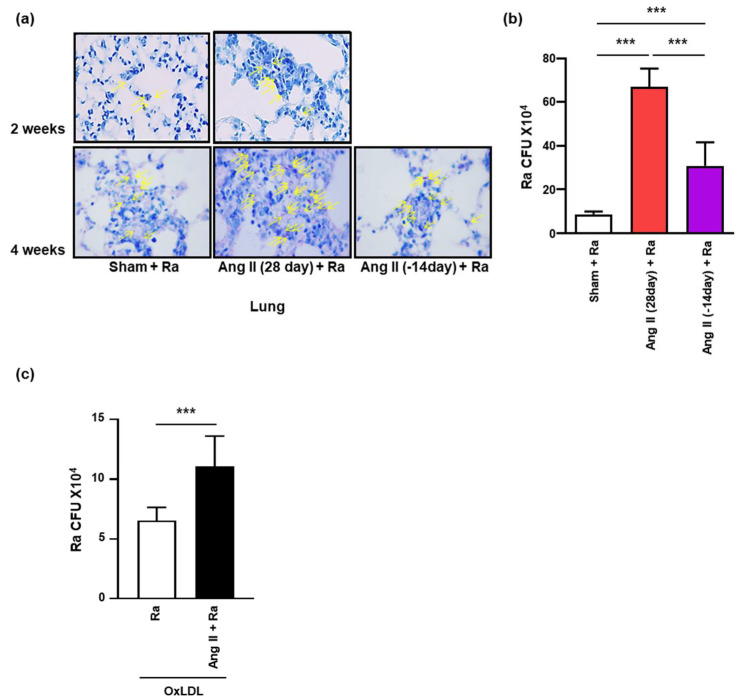
Ang II-induced hypertensive condition promotes the intracellular survival of myocbacteria. Hypertension was induced in C57BL/6 mice by Ang II (0.7 mg/kg/day) infusion using an osmotic mini-pump for 14 or 28 days. PBS was used in the control group. Mice were infected with Mtb H37Ra (1 × 10^6^ CFU) by intratracheal injection at after 1 week of Ang II infusion. (**a**) Acid-fast bacillus (arrows) staining was performed in lung tissues of Mtb H37Ra (1 × 10^6^ CFU)-infected mice after 2 or 4 weeks of Ang II infusion (magnification, 40×). Data are presented as representative pictures of each group. (**b**) The total number of Mtb H37Ra was counted by CFU analysis in the lungs of five infected mice in each group. (**c**) BMDM cells were pretreated with Ang II (1 μM) for 1 h followed by incubation with 50 μg/mL OxLDL after Mtb H37Ra (MOI = 1) infection. Intracellular bacterial survival was measured by enumeration of CFU at 48 h after infection. Data are shown as means ± SD (*** *p* < 0.001).

**Figure 7 cells-10-02478-f007:**
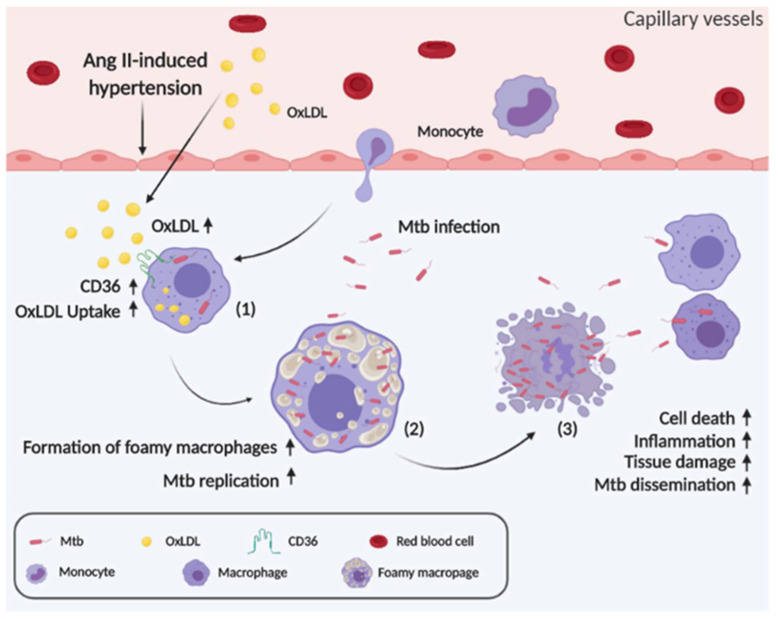
Schematic diagram illustrating the role of Ang II-induced hypertension during Mtb infection. (1) Ang II-induced hypertension increases CD36 expression on the surface of macrophages. CD36 expression is required for the accumulation of OxLDL in Mtb infected macrophages. (2) Lipid-loaded macrophages facilitate the replication of Mtb. (3) Formation of foamy macrophages promotes increased cell death and tissue damage, leading to dissemination of Mtb.

## Data Availability

The datasets supporting the conclusions of this article are included within the article and its additional files.

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
