# Peer review of "Ang II-Induced Hypertension Exacerbates the Pathogenesis of Tuberculosis"

_cells, 2021, doi:10.3390/cells10092478_

Round 1

Reviewer 1 Report

The article describes the role of Ang II in the pathogenesis of tuberculosis and is well written.

There are minor points which are as follows

Comments:

Page 1, line 30: (ref needed)

Page 3, line 116: LDH is not defined when used here and it seems it is defined on page 6

Author Response

Reviewer 1.

The article describes the role of Ang II in the pathogenesis of tuberculosis and is well written.

There are minor points which are as follows

Comment 1) Page 1, line 30: (ref needed)

→ A: Thank you for your comment, we added the reference in Page 1.

Comment 2) Page 3, line 116: LDH is not defined when used here and it seems it is defined on page 6.

→ A: We revised it in page 3 and page 6 as follows.

“ Hypoxia induced factor 1α (HIF-1α, 1:100; Santa Cruz Biotechnology, Santa Cruz, CA), and lactate dehydrogenase A (LDH-A, 1:100; Abcam, Cambridge, MA) using an LSAB2 system horseradish peroxidase (HRP) kit (Dako, Carpinteria, CA).” in page 3.

“Next, we compared the production of LDH-A in mouse lungs because measuring LDH-A release is a useful method for detection of necrosis” In page 6.

Reviewer 2 Report

This work describes the possible involvement of angiotensin II-induced hypertension in the worsening of lung pathology in the course of tuberculosis infection, as well as facilitating Mtb survival. This subject is novel and relevant for the study of tuberculosis.

Major comments:

The rationale presented in the introduction section is confuse. The authors associate atherosclerosis and cardiovascular disease to bacterial infections and cell death to justify a possible role of angiotensin II in the intracellular survival of Mycobacterium tuberculosis. I believe it should be thoroughly reviewed.

The mouse model of hypertension using osmotic minipumps for angiotensin II infusion that I have found referred in the literature involves the infusion of 50 to 1500 ng x kg-1 x min-1 of angiotensin II (1-4). The authors state that they have used 0.7 mg∙kg−1∙day−1. This dose seems very discrepant compared to these other studies. Please review this information and provide a reference of this model if it is correct. 

In Figs. 2 and 3, Cas3 and LDH-A levels in sham treated Ra infected mice appear to be higher than those found in infected mice treated for two weeks with angiotensin II. The authors do not comment these findings nor provide the p-values for these comparisons.

Minor comments

The error bars are not displayed in the graph of Fig. 1a.

In the legend of Figure 1, regarding Fig. 1b, the authors provide the following information: "Data shown are the means ± SEM of five mice in each group. Data are presented as the means ± SD from three mice in each group. Data are representative of three independent experiments (††† p < 0.001, *** p < 0.001)." It is not clear whether a second graph would be depicted in this figure.

On lines 239-241, the authors state that "Next, we investigated whether high blood pressure affects the death of Mtb-infected macrophages because our group has discovered that apoptosis plays an important role in the destruction of tuberculosis bacteria in macrophages [29]." The direct effects of high blood pressure on Mtb-infected macrophages were not investigated in this work. Also, Ref 29 exploits the role of ER stress response via CHOP induction in cultures infected with Mtb H37Rv, relating this particular pathway to apoptosis and Mtb survival. Please review this text and provide references that address the role of apoptosis in Mtb infection in a more comprehensive way, or provide evidence that the mechanism underlying angiotensin II-driven apoptosis may be related to ER stress.

In Fig. 5, the differences shown in necrotic cell death and LDH release comparing cultures infected with Ra with or without angiotensin II pre-treatment are significant, but very small. The biological significance of these findings should be discussed?

In lines 361-362, the authors state that "Next, we investigated whether Ang II-induced hypertension increases the dissemination and survival of Mtb in foamy macrophages". It is not clear how the authors addressed the possible role of angiotensin II in the dissemination of Mtb.

In lines 392-394, the authors state that "Our results showed that Ang II stimulation increased CD36 expression and OxLDL uptake in macrophages, resulting in enhanced intracellular survival of mycobacteria (Figure. 7)." It is not possible to infer this causal relationship from the results presented in this work. 

  1. Tsukamoto Y, Mano T, Sakata Y, Ohtani T, Takeda Y, Tamaki S, et al. A novel heart failure mice model of hypertensive heart disease by angiotensin II infusion, nephrectomy, and salt loading. Am J Physiol-Heart Circ Physiol. 2013 Dec 1;305(11):H1658–67.
  2. Gomolak JR, Didion SP. Angiotensin II-induced endothelial dysfunction is temporally linked with increases in interleukin-6 and vascular macrophage accumulation. Front Physiol. 2014;5:396.
  3. Crowley SD, Gurley SB, Herrera MJ, Ruiz P, Griffiths R, Kumar AP, et al. Angiotensin II causes hypertension and cardiac hypertrophy through its receptors in the kidney. Proc Natl Acad Sci. 2006 Nov 21;103(47):17985–90.
  4. Zimmerman MC, Lazartigues E, Sharma RV, Davisson RL. Hypertension caused by angiotensin II infusion involves increased superoxide production in the central nervous system. Circ Res. 2004 Jul 23;95(2):210–6.

Author Response

 Reviewer 2

Comments and Suggestions for Authors

This work describes the possible involvement of angiotensin II-induced hypertension in the worsening of lung pathology in the course of tuberculosis infection, as well as facilitating Mtb survival. This subject is novel and relevant for the study of tuberculosis.

Major comments:

Comment 1) The rationale presented in the introduction section is confuse. The authors associate atherosclerosis and cardiovascular disease to bacterial infections and cell death to justify a possible role of angiotensin II in the intracellular survival of Mycobacterium tuberculosis. I believe it should be thoroughly reviewed.

→ A: Thank you for your valuable comments. It has been reported that bacterial infection is associated with cardiovascular disease and atherosclerosis (Khademi F et al., Arch Med Sci 2019. 15(4): 902-911; Campbell LA et al., Arch Med Res 2015. 46(5): 339-350). A previous report suggested that risk of cardiovascular disease in persons who develop tuberculosis is higher than in persons without a history of tuberculosis (Huaman MA et al., Trop Dis Travel Med Vaccines 2015. 1:10). A recent paper showed that tuberculosis appears to be a marker for increased CVD risk (Basham CA et al., PLoS One 2020. 15(7): e0235821). Chronic infection with several viruses and bacteria has been reported in association with high blood pressure (Vahdat K et al., Am J Hypertens 2013. 26(9): 1140-1147). Interestingly, hypertension is reported to be associated with the severity and fatality of SARS-CoV-2 infection (Zhang J et al., Epidemiol Infect 2020. 148: e106).   

Tuberculous granulomas in the lung are organized with infected macrophages in the center surrounded by lymphocytes and the macrophages can differentiate into the lipid loaded foamy macrophages [Peyron P et al., PLoS Pathog 2008. 4(11): e1000204]. The accumulation of lipid bodies in macrophages can lead to nutrient-rich niches and continuous replication of Mtb and dormant state [Peyron P et al., PLoS Pathog 2008. 4(11): e1000204; Lee W et al., J Biol Chem 2013. 288(10): 6788-800]. Atherosclerosis contributes to foamy macrophage formation (Moore K et al., Nat Rev Immunol 2013. 13(10): 709-721; Yu XH et al., Clinica Chimica Acta 2013 424(23):245-252). Since it is well known that hypertension is not only a well-established cardiovascular risk factor but also increases the risk of atherosclerosis (Alexander RW. Hypertension 1995. 25: 155-161), we have tried to reveal the relationship between hypertension and severe pathogenesis of tuberculosis in this study. Therefore, it is reasonable to hypothesize that AngII-induced hypertension will affect Mycobacterium tuberculosis survival.

Comment 2) The mouse model of hypertension using osmotic minipumps for angiotensin II infusion that I have found referred in the literature involves the infusion of 50 to 1500 ng x kg-1 x min-1 of angiotensin II (1-4). The authors state that they have used 0.7 mg∙kg−1∙day−1. This dose seems very discrepant compared to these other studies. Please review this information and provide a reference of this model if it is correct.

→ A: The concentration of AngII is within the range that you suggest.

50 to 1500 ng x kg-1 x min-1

  • 50ng*60min*24h=72.000ng/kg/day (=0.072mg/kg/day),
  • 1500ng*60min*24=2160000ng/kg/day (=2.16mg/kg/day)

Various doses of Ang II have been used in many studies. We used the concentration of Ang II based on several previous reports. In our study, the diastolic blood pressure of the mice is effectively increased by AngII (0.7 mg∙kg−1∙day−1). We have provided references for your understanding.

  1. E. Garza, et al. Striatin heterozygous mice are more sensitive to aldosterone-induced injury. J Endocrinol 2020;245(3):439-450
  2. Guivarc'h, et al. Nuclear Activation Function 2 Estrogen Receptor alpha Attenuates Arterial and Renal Alterations Due to Aging and Hypertension in Female Mice. J Am Heart Assoc 2020;9(5):e013895
  3. Feng, et al. Paired box 6 inhibits cardiac fibroblast differentiation. Biochem Biophys Res Commun 2020;528(3):561-566
  4. N. Dinh, et al. Aldosterone-Induced Hypertension is Sex-Dependent, Mediated by T Cells and Sensitive to GPER Activation. Cardiovasc Res 2020;

Comment 3) In Figs. 2 and 3, Cas3 and LDH-A levels in sham treated Ra infected mice appear to be higher than those found in infected mice treated for two weeks with angiotensin II. The authors do not comment these findings nor provide the p-values for these comparisons.

→ A: We revised it as follows; We changed " The production of cleaved Cas-3 was elevated in lung tissue of Ang II-induced hypertensive mice comparing to control during Mtb infection." to "The production of cleaved Cas-3 was elevated in lung tissue of Mtb infected mice comparing to control mice. Ang II -induced hypertension more enhances the cleaved Cas-3 than control during Mtb infection." in page 6.

Minor comments

Comment 1) The error bars are not displayed in the graph of Fig. 1a.

In the legend of Figure 1, regarding Fig. 1b, the authors provide the following information: "Data shown are the means ± SEM of five mice in each group. Data are presented as the means ± SD from three mice in each group. Data are representative of three independent experiments (††† p < 0.001, *** p < 0.001)." It is not clear whether a second graph would be depicted in this figure.

→ A: We revised the part you pointed out in fig 1 legend (page 6).

Comment 2) On lines 239-241, the authors state that "Next, we investigated whether high blood pressure affects the death of Mtb-infected macrophages because our group has discovered that apoptosis plays an important role in the destruction of tuberculosis bacteria in macrophages [29]." The direct effects of high blood pressure on Mtb-infected macrophages were not investigated in this work. Also, Ref 29 exploits the role of ER stress response via CHOP induction in cultures infected with Mtb H37Rv, relating this particular pathway to apoptosis and Mtb survival. Please review this text and provide references that address the role of apoptosis in Mtb infection in a more comprehensive way, or provide evidence that the mechanism underlying angiotensin II-driven apoptosis may be related to ER stress.

→ A: Thank you for your comments, we revised the text and added proper references as follows.

“Previous reports suggeted that ER stress plays an important role in Mtb infected macrophages and ER stress contributes to the pathogenesis of hypertension [Lee J et al., Antioxidants 2021 10(6):872; Lim YJ, Choi JA, Choi HH, Cho SN, Kim HJ, Jo EK, Park JK, Song CH. PLoS One 2011, 6(12):e28531; Colin N Young et al, Exp Physiol.2017 Aug 1;102(8):869-884.].”

Comment 3) In Fig. 5, the differences shown in necrotic cell death and LDH release comparing cultures infected with Ra with or without angiotensin II pre-treatment are significant, but very small. The biological significance of these findings should be discussed?

→ A: Although LDH release seems to be a small difference, the difference is statistically significant. Anyway, we revised the discussion as follows;

It is well known that necrosis helps in dissemination of Mtb (Behar, S. M. et al. Mucosal Immunol 2011. 4: 279-287; Martin, C. J. et al. Cell Host Microbe 2012. 12:289-300.)

Comment 4) In lines 361-362, the authors state that "Next, we investigated whether Ang II-induced hypertension increases the dissemination and survival of Mtb in foamy macrophages". It is not clear how the authors addressed the possible role of angiotensin II in the dissemination of Mtb.

→ A: We think that Ang II-induced hypertension might affect foamy macrophage formation. Previous reports showed that foamy macrophages contribute to survival and maintenance of Mtb (Russell DG et al., Nat Immunol. 2010 10(9): 943-948; Dahee Shim et al, Front Immunol. 2020. 11:910). Another report suggested that bacterial infection is associated with cardiovascular disease and atherosclerosis (Khademi F et al., Arch Med Sci 2019. 15(4): 902-911; Campbell LA et al., Arch Med Res 2015. 46(5): 339-350). Therefore, Ang II-mediated hypertension can affect the formation of macrophages, providing an environment in which mycobacteria can survive.

We revised the text in page 12. “Previous reports suggested that foamy macrophages give an excuse for survival and maintenance of Mtb (Dahee Shim et al, Front Immunol. 2020 May 12;11:910. PMID32477367).

Comment 5) In lines 392-394, the authors state that "Our results showed that Ang II stimulation increased CD36 expression and OxLDL uptake in macrophages, resulting in enhanced intracellular survival of mycobacteria (Figure. 7)." It is not possible to infer this causal relationship from the results presented in this work.

 → A: Thank you for your comment. We revised the text "Our results showed that Ang II stimulation increased CD36 expression and OxLDL uptake in macrophages, resulting in enhanced intracellular survival of mycobacteria (Figure. 7)” to "Our results suggested that Ang II stimulation increased CD36 expression and OxLDL uptake in macrophages, resulting in enhanced intracellular survival of mycobacteria (Figure. 7). " in page 13.